# A Comparative Assessment of the Baking Quality of Hybrid and Population Wheat Cultivars

**Marta Jańczak-Pieniążek [1],\***, **Jan Buczek [1]**, **Joanna Kaszuba [2]**, **Ewa Szpunar-Krok [1]**, **Dorota Bobrecka-Jamro [1]** and **Grażyna Jaworska [2]**

[1]   Department of Crop Production, University of Rzeszow, Zelwerowicza 4 St., 35-601 Rzeszow, Poland; jbuczek@ur.edu.pl (J.B.); szpunar-krok@wp.pl (E.S.-K.); bobjamro@ur.edu.pl (D.B.-J.)

[2]   Department of Food Technology and Human Nutrition, University of Rzeszow, Zelwerowicza 4 St., 35-601 Rzeszow, Poland; jkaszuba@ur.edu.pl (J.K.); rrgjawor@cyf-kr.edu.pl (G.J.)

\*   Correspondence: mjanczak@ur.edu.pl

**Abstract:** The study assessed the quality parameters of grain and flour, the rheological properties of dough and the quality of bread prepared from flour of hybrid cultivars of wheat in comparison with population cultivars of wheat. As the interest in wheat hybrids cultivars from the agricultural and milling industry is growing, their technological value of grain and flour was evaluated at two levels of nitrogen fertilisation (N1—110 kg/ha, N2—150 kg/ha). Increasing the fertilisation (N2) produced a significant influence on the crude protein and gluten content in the flour, as well as the moisture of the crumb and the yield of the dough without impacting other rheological traits and parameters of bread baking process. The performed principal component analysis (PCA) allowed for identification of the best cultivars among the studied wheat cultivars (Hybery and Hyvento). The hybrid cultivar Hyvento was characterised by favourable qualitative traits of the grain (vitreousness, crude protein content) and rheological parameters of the dough (bread volume), however, it had lower baking quality parameters. Among the hybrid cultivars, the best applicability for baking purposes was Hybery due to the favourable values of the baking process parameters and bread quality (bread yield, bread volume, Dallmann porosity index of crumb). Hybrid cultivars of wheat can therefore be used for the production of bread and be an alternative in agricultural production for population cultivars, which will contribute to filling the knowledge gap for the hybrid wheat cultivars.

**Keywords:** hybrid wheat; rheological properties; bread-making quality; N fertilisation

## 1. Introduction

Wheat (*Triticum aestivum* L.) production for food purposes in Europe and around the world dictates the consumption model and determines the safety of food and human nutrition [1,2].

With its high content of complex carbohydrates, wheat grain is a significant source of the energy provided in food, particularly due to the high contribution of cereal products (flour, groats, flakes, pasta, bread) in the daily diet of people. What is more, it constitutes a basic and valuable source of protein, minerals (P, K, Ca, Mg and micronutrients), B group vitamins, dietary fibre, and antioxidants, being at the same time one of the major allergens [3,4].

In combination with water, wheat flour creates dough with unique viscoelastic properties (distinguishing it among the flours of other cereals), enabling processing the dough for bread, pasta and other food products, which stems from the presence of gluten proteins in its composition [5,6].

Baking quality of wheat flour depends largely on the amount of seed storage proteins, in particular glutenins, the content of which is dictated by, among other things, the climate and soil conditions and fertilisation, in particular with nitrogen, which is also a cultivar-related trait [7,8]. Thus, the increased

glutenin content contributes to increased dough resistance to stretching and prolonged dough development time, whereas a higher HMW glutenin content increases dough resistance and bread volume [9].

What is more, every new cultivar possesses its own set of genes controlling the synthesis of gluten proteins and affecting the formation of a different quality of gluten, determining the specified value of quality traits of the wheat grain and flour. Nowadays, as a result of accelerated biological advances, attempts are being made to obtain wheat cultivars with more favourable grain quality properties enriched in terms of nutrition with the greatest possible amount of bioactive substances, while being tolerant of stress and variable climatic and soil conditions [10].

One strategy in wheat breeding is the improvement of yield stability and size under stress conditions created by biotic and abiotic conditions of the environment by means of hybrid vigour [11]. Interest of the public and private sector in hybrid cultivars of wheat has been increasing, not only in the countries with growing concerns about food safety (China, India, Pakistan), but also in Europe and the USA [2].

The surface area occupied by hybrid wheat is less than 1% of the global wheat cultivation area [1,12]. In Europe, hybrid wheat is cultivated on an area of over 560.000 ha, and 80% of hybrid cultivars are grown in France, while the remaining 20% are grown in Germany, Hungary, Italy, the Czech Republic, Slovakia, Romania and Portugal [2]. In hybrid wheat, the vigour obtained via cross-fertilisation differs genetically from the distinct parent lines, resulting in an improved phenotype and traits desired for plant production, such as: higher rate of growth and plant differentiation, higher and stable grain yield, better resistance to environmental stress conditions and greater competitiveness with regards to diseases and weeds [11,12].

Thus far, the literature data concerning the assessment of the technological value of grain and flour from hybrid winter wheat cultivars is scarce. The objective of the conducted study was assessment of the quality of grain and flour, baking quality and applicability for the production of bread of flour obtained from the grain of hybrid wheat cultivars as compared with population wheat cultivars. The wheat grain originated from an experiment utilising two levels of nitrogen fertilisation.

## 2. Materials and Methods

### 2.1. Experimental Material

The study material consisted of winter wheat grain (*Triticum aestivum* L.) from seven hybrid cultivars: Hybery, Hyena, Hyfi, Hyking, Hymalaya, Hypocamp, and Hyvento. The wheat grain of two population cultivars, Belissa and Hondia were used as control samples. The wheat grain originated from an experiment utilising two levels of nitrogen fertilisation ($N_1$—110 kg/ha, $N_2$—150 kg/ha), carried out in the 2017/2018 period at the Experimental Station for Cultivar Assessment in Przecław (50°11′ N, 21°29′ E, altitude 185 m asl) near Mielec (Poland). The doses of nitrogen fertilisation were applied according to the COBORU method [13] intended for strict field experiments with winter wheat cultivars, at two levels of agricultural technology: medium intensity (A1) with $N_1$ fertilisation—110 kg/ha and high intensity (A2) with $N_2$ fertilisation—150 kg/ha. From the grain of the investigated cultivars flour, was obtained, which was subject to testing in the field of baking quality assessment.

### 2.2. Methods

#### 2.2.1. Grain Quality Assessment

Grain samples were analysed for bulk density, referred to as mass per hectolitre, by means of EN ISO 7971-3:2019 [14]. Thousand grain weight (TGW) was determined with a grain counter (Sadkiewicz Instruments, Poland). Vitreousness was determined according to ICC Standard Method No. 129. [15]. Vitreousness was determined by cutting 50 grains with a Farinotom (Sadkiewicz Instruments, Poland)

and calculating the number of grains that have completely or partially vitreous flour on the cross-section surface, measured as a percentage of vitreous kernels (0–100%).

### 2.2.2. Grain Milling

Grain with a moisture level of $13.0 \pm 0.1\%$ was milled in a Quadrumat Junior mill (Brabender, Germany) with a cone screen, mesh size $\phi = 212\ \mu m$ in accordance with AACC Method No. 26-50.01 [16]. Flour yield [%] was calculated as the amount of flour obtained via milling 100 g of the grain.

### 2.2.3. Grain and Flour Quality Assessment

Total ash was determined in accordance with ICC Standard Method No. 104/1 [17]. Nitrogen content was measured and calculated into crude protein content using the $N \times 6.25$ conversion ratio based on AACC Method No. 46-11.02 [16]. Wet gluten and dry gluten content and gluten index (GI) were determined in the flour following AACC Method No. 38-12.02 [16] using a Gluten Index System device (Perten, Sweden). Finally, the flour falling number was determined with a Falling Number 1800 apparatus (Perten, Sweden) following AACC Method No. 56-81.03 [16].

### 2.2.4. Farinograph Parameters Testing

Water absorption of the flour at a maximum consistency of 500 FU (Farinograph Units) as well as the development time of the dough and dough stability, degree of softening and quality number were determined using a Farinograph-E (Brabender, Germany) following ICC Standard No. 115/1 [17].

### 2.2.5. Bread Baking

The baking procedure was carried out with a modified ICC Standard Method No. 131 [17]. The dough was prepared from 300 g flour, 9 g yeast, 4.5 g salt and water, allowing for production of dough with a consistency equal to 350 FU, determined with a Farinograph-E (Brabender, Germany). The ingredients were combined with the use of a laboratory mixer (Sadkiewicz Instruments, Poland) and afterwards the dough was kneaded by hands for 3 min. The dough was placed in a fermentation chamber under conditions of 80% humidity and 30 °C (Sveba Dahlen, Sweden) for 30 min, then it was kneaded by hands for 3 min and again placed in the fermentation chamber for another 30 min. Subsequently, 250 g pieces of dough were formed, which were placed in greased baking pans. Fermentation of the dough pieces was continued until their optimum rise in the fermentation chamber was obtained (as above). The dough pieces were baked in a Classic electric oven (Sveba Dahlen, Sweden) at a temperature of 230 °C for 30 min. The breads were weighed twice: right after removing from the oven and after cooling down.

### 2.2.6. Bread Quality Assessment

Weight and volume were measured approximately 24 h from baking. The basic indicators of the wheat bread baking process, such as dough yield, total baking loss, and bread yield were calculated [18]. Volume of 100 g bread was calculated after weighing and measuring the loaves with a Sa-Wa volumeter (Sadkiewicz Instruments, Poland) following AACC Method No. 10-05.01 [16]. The Dallmann porosity index of the crumb was determined according to the Dallmann scale [19], in which 30 points are given to the lowest quality and 100 to the highest quality crumb. The bread's crumb moisture was determined according to AACC Method No. 44-15.02 [16].

### *2.3. Statistical Analysis of Results*

The results of three replicate analyses are presented as mean values ± standard deviations and the coefficient of variation (CV). The two-way ANOVA was used to analyse the data. The Duncan test was used to determine the statistically significant difference at the 95% level ($p = 0.05$). In addition, the Pearson's linear correlation coefficients were calculated between the analysed parameters describing

grain, flour and bread at the significance level of $p = 0.05$. Additionally, the principal component analysis (PCA) was used to provide a ready means of visualizing the differences and similarities between the investigated wheat cultivars in different nitrogen fertilisation. Statistical analysis of the results was performed using TIBCO Statistica 13.3.0 (TIBCO Software Inc., Palo Alto, CA, USA).

## 3. Results and Discussion

### 3.1. Grain Quality and Flour Yield

Table 1 lists the results of the assessment of selected quality parameters of winter wheat grain. According to Morgan et al. [20] TGW is linked to the yield and quality of the obtained flour, and determines its colour and ash content. No significant impact of increasing the nitrogen dose from $N_1$ to $N_2$ on the TGW change could be found in the present study, for hybrid as well as population cultivars. The lowest value of the parameter was determined forHyking cultivar ($N_1$—38.6 g, $N_2$—41.6 g). The population Hondia cultivar and the Hypocamp hybrid cultivar fertilised with $N_2$ nitrogen dose exhibited TGW values higher by 17.8 and 14.4% in comparison to the Hyking hybrid cultivar. Klikocka et al. [21] were also unable to determine the impact of different nitrogen fertilisation levels on TGW formation, whereas Abedi et al. [22] demonstrated a significant increase of the values of this parameter values under the impact of elevated nitrogen doses.

**Table 1.** Selected qualitative indicators of wheat grain and flour from hybrid and population cultivars.

| Cultivar | Nitrogen | TGW [g] | Bulk Density [kg/hl] | Vitreousness [%] | Crude Protein Content [%] | Flour Yield [%] |
|---|---|---|---|---|---|---|
| Hybery | $N_1$ | 43.6 [b,c] ± 2.5 | 74.5 [b–d] ± 0.7 | 44 [a,b] ± 9 | 11.8 [a] ± 0.4 | 75.3 [b] ± 0.8 |
| | $N_2$ | 46.5 [c–e] ± 2.1 | 76.0 [d–f] ± 1.6 | 48 [a,b] ± 0 | 12.7 [b,c] ± 0.4 | 75.7 [b,c] ± 0.3 |
| Hyena | $N_1$ | 44.6 [b,c] ± 1.3 | 72.2 [a,b] ± 0.4 | 54 [a–c] ± 6 | 11.6 [a] ± 0.1 | 77.9 [e–g] ± 0.9 |
| | $N_2$ | 45.1 [c,d] ± 0.3 | 75.3 [c–f] ± 0.9 | 77 [d] ± 6 | 13.0 [c,d] ± 0.1 | 78.7 [g,h] ± 0.6 |
| Hyfi | $N_1$ | 44.1 [b,c] ± 3.1 | 74.3 [b–d] ± 1.6 | 40 [a] ± 1 | 12.7 [b,c] ± 0.4 | 78.2 [f–h] ± 0.6 |
| | $N_2$ | 46.9 [c–e] ± 0.4 | 73.1 [a–c] ± 0.4 | 53 [a,b] ± 2 | 14.1 [f] ± 0.1 | 78.5 [g,h] ± 0.4 |
| Hyking | $N_1$ | 38.6 [a] ± 2.3 | 71.1 [a] ± 1.6 | 50 [a,b] ± 7 | 12.6 [b,c] ± 0.1 | 75.6 [b,c] ± 0.4 |
| | $N_2$ | 41.6 [a,b] ± 1.9 | 74.6 [b–d] ± 0.4 | 75 [d] ± 8 | 13.7 [e,f] ± 0.3 | 77.5 [e–g] ± 0.1 |
| Hymalaya | $N_1$ | 45.6 [c,d] ± 1.2 | 74.8 [b–e] ± 0.1 | 71 [d] ± 5 | 12.3 [b] ± 0.2 | 80.1 [I] ± 0.6 |
| | $N_2$ | 46.1 [c,d] ± 1.5 | 77.1 [e,f] ± 0.7 | 75 [d] ± 7 | 13.5 [d,e] ± 0.1 | 79.5 [h,i] ± 0.0 |
| Hypocamp | $N_1$ | 47.2 [c–e] ± 0.6 | 77.6 [f] ± 0.4 | 45 [a,b] ± 1 | 11.7 [a] ± 0.2 | 76.8 [c–e] ± 0.1 |
| | $N_2$ | 48.6 [d–f] ± 0.6 | 77.6 [f] ± 1.5 | 51 [a,b] ± 10 | 12.8 [b,c] ± 0.1 | 81.9 [j] ± 0.5 |
| Hyvento | $N_1$ | 46.0 [c,d] ± 2.0 | 75.4 [c–f] ± 2.3 | 67 [c,d] ± 4 | 12.4 [b] ± 0.3 | 77.7 [e–g] ± 0.8 |
| | $N_2$ | 47.1 [c–e] ± 1.3 | 73.8 [b–d] ± 1.6 | 74 [d] ± 1 | 14.2 [f] ± 0.3 | 78.8 [g,h] ± 0.4 |
| Belissa [1] | $N_1$ | 43.9 [b,c] ± 0.0 | 73.6 [a–d] ± 0.2 | 45 [a,b] ± 5 | 11.4 [a] ± 0.3 | 70.8 [a] ± 0.7 |
| | $N_2$ | 46.9 [c–e] ± 0.3 | 74.3 [b–d] ± 0.4 | 56 [b,c] ± 10 | 12.7 [b,c] ± 0.2 | 75.1 [d–f] ± 0.9 |
| Hondia [1] | $N_1$ | 49.9 [e,f] ± 0.1 | 73.3 [a–c] ± 0.3 | 50 [a,b] ± 7 | 12.5 [b] ± 0.2 | 74.7 [c–e] ± 0.3 |
| | $N_2$ | 50.6 [f] ± 0.0 | 74.0 [b–d] ± 1.2 | 57 [b,c] ± 4 | 13.3 [d,e] ± 0.1 | 75.2 [b–d] ± 0.5 |
| CV (%) ** | $N_1$ | 7.23 | 2.68 | 21.56 | 4.20 | 3.45 |
| | $N_2$ | 5.45 | 2.26 | 20.00 | 4.39 | 3.76 |

[1] Population cultivars. The results are presented as mean values ± standard deviation. Different letters in the same column indicate significant differences ($p = 0.05$), according to ANOVA followed by Duncan test. ** CV coefficient of variation; $N_1$—110 kg/ha. $N_2$—150 kg/ha.

The population cultivars significantly differed from the hybrids at both nitrogen treatments only in the case of the Hypocampcultivar. The bulk density is an important quality indicator of grain, determining its accuracy, the degree of grain development, their structure, and the thickness of the coat, and it also determines the milling value of the grain. However, bulk density is not always an ideal measure of grain quality, because this parameter is determined by environmental factors. A higher bulk density indicates a better technological value of the wheat, because smaller grains can have a much lower ratio of endosperm to coat and germ [23]. The highest grain bulk density value was found for the Hypocamp hybrid cultivar (77.6 kg/hl), without the impact of $N_1$ and $N_2$ doses on the value of this trait.

An increasing tendency of bulk density was observed as a result of applying $N_2$ fertilisation. However, the Hypocamp hybrid cultivar was characterised by a higher value of bulk density in $N_1$ fertilisation than the Hyfi, Hyking, Hyvento, Hondia, and Belissa cultivars with $N_2$ fertilisation. Among hybrid cultivars, only Hyena and Hyking were characterised by a significant increase of grain bulk density after the application of the $N_2$ dose [22]. Harasim and Wesołowski [24] indicated that winter wheat grain, as a result of increasing the nitrogen fertilisation dose from 100 to 150 kg/ha, was characterised by a higher value of the analysed parameter. According to Dziki et al. [25] vitreousness grains are characterised by a higher degree of endosperm content, and higher grain hardness and protein content in comparison with non-vitreous grain. The grain vitreousness of the tested cultivars ranged between 40.0 and 77.0%. Significantly, the highest grain vitreousness for both doses: $N_1$ and $N_2$ were found in the Hymalaya and Hyventohybrid cultivars, which at $N_1$ fertilisation, were characterised by a higher value of this parameter than the Hybery, Hyfi, Hypocamp, Belissa, and Hondia cultivars fertilised by $N_2$. Compared to $N_1$, $N_2$ resulted in a significant increase of grain vitreousness, by 23.0 and 25.0% only in the Hyena and Hykinghybrid cultivars. An appropriate content of total protein, which is not only determined by nitrogen fertilisation, but also by genetic predispositions, is the basic factor determining the applicability of wheat cultivar grain for bread baking [10,26,27]. Crude protein content in the grain ranged between 11.8 and 14.2% (hybrid cultivars) and between 11.4 and 13.3% (population cultivars). Similarly to a study by Skudra and Ruza [3], the present study confirmed increased protein content in the grain with increased N dosage for all cultivars. However, the cultivars reacted differently to this effect [9]. The Hyfi and Hyking cultivars fertilised with $N_1$ had a crude protein content at the same level as the Hybery, Hypocamp and Belissa cultivars fertilised with $N_2$. Also Haque et al. [28] indicated the effect of the interaction of wheat cultivars and different levels of N fertilisation on protein content of the grain. Increasing the dose from $N_1$ to $N_2$ resulted in a significantly higher crude protein content in the grain of the Hyena and Hykinghybrid cultivars, at the level determined in the grain of the Hondiapopulation cultivar. The highest crude protein content ($N_2$) was determined in the Hyfi and Hyvento hybrid cultivars [10,26]. The grain of the hybrid cultivars was generally characterised by a higher flour yield than the population cultivars. The highest flour yield was obtained from the grain of the Hymalaya ($N_1$) and Hypocamp ($N_2$) hybrid cultivars. Increasing the fertilisation dosage to $N_2$ had a significant impact on increasing the flour yield in Hypocamp and Hyking hybrid cultivars and the Belissa population cultivar, by 9.5, 2.5, and 8.2%, respectively. A similar wheat flour yield without the impact of fertilisation variants was obtained by Warechowska et al. [29], whereas Metho et al. [30] indicated a significant interaction between cultivars and nitrogen fertilisation on the value of the flour yield. In this study it was shown that the Hymalaya hybrid cultivar fertilised with $N_1$ was characterised by higher flour yield than other cultivars fertilised with $N_2$ with the exception of Hypocamp.

The Belissa population cultivar ($N_1$, $N_2$) is concerned, the Hypocamp hybrid cultivar developed a grain with a lower TGW and vitreousness but with a higher bulk density, and the crude protein content in the grain of both cultivars was similar. Especially, for the Hypocamp hybrid cultivar grain with the $N_2$ dose, a higher flour yield was obtained than from the grains of the Belissa and Hondia population cultivars.

Among the analysed features, the lowest variability was found in bulk density and simultaneously the highest grain vitreousness.

### 3.2. Flour Quality Parameters

Table 2 presents selected parameters of the quality of the flour obtained from the grain of the tested wheat cultivars. The ash content in the grain depends mainly on the genotype, wheat class, and cultivar as well as the growing location, year, and grinding method [31,32]. The ash content in the flour of the hybrid cultivars ranged from 0.43 to 0.69%. Compared to the population cultivars and other hybrid cultivars, the flour of the Hypocamp and Hybery hybrids was characterised by the lowest ash content. The highest ash content was found in the flour of the Hyfi (0.59–0.73%) and Hyking (0.69–0.58%)

hybrid cultivars. According to Bucsella et al. [33] and Hemery et al. [34], an increase in the ash content in the flour combined with an increase in nutrients (fibre, vitamins) is desirable, but the technology quality of the flour is then lower due to weakening of the protein matrix during dough formation. Increased N fertilisation resulted in a reduced total ash content in the flour of Hyking hybrid cultivar (by 15.9%)), and an increase for Hyfi cultivar (by 19.2%) and Hondia population cultivar (by 20.7%). A study by Bayoumi and El-Demardash [35] also indicated a beneficial effect on this feature of nitrogen fertilisation, although in a later study by Warechowska et al. [29], such a relationship was not found. Statistical analysis revealed a negative linear correlation between total ash content in flour and bulk density (r = −0.62, $p < 0.05$). The falling number indicates of activity of amylolytic enzymes contained in the flour, their capacity for hydrolysing the starch present in the flour to sugars, which are substrates for the dough fermentation process [10,36]. A study by of Linina and Ruza [37] shows that the falling number value depends significantly on the weather conditions, grain storage time and the applied nitrogen dose. In the present study, the increased N dose resulted in a reduced flour falling number (by approximately 20 s) in the Hyfi, Hybery, and Hymalaya hybrid cultivars and Hondiapopulation cultivar. Kindred et al. [36] demonstrated that the impact of nitrogen fertilisation on the falling number value relies on the interactions between the genotype and N doses, which has been confirmed in the present study. The application of the $N_2$ dose increased the value of the falling number in the Belissa population cultivar (by 3.5%) and in the hybrid cultivars in the range from 0.3 (Hyking Hypocamp) to 1.6% (Hyvento). Most flours used in the baking industry requires adjustment of the falling number to the perfect range (250–320 s), and a reduced value of this parameter is obtained by adding $\alpha$-amylase preparation [38]. In the presented study, this condition was fulfilled by flours of hybrid wheat cultivars, with the exception of the Hyena and Hyventocultivars and the Belissa and Hondia population cultivars. Among the hybrid cultivars, the lowest falling number value from 190 ($N_1$) to 210 s ($N_2$) was found in the Hyfihybrid cultivar and the highest from 387 ($N_2$) to 389 s ($N_1$) Hyena cultivar. Jaskulska et al. [8] state that wheat cultivars react in a natural manner to the N fertilisation level, and it is displayed by grain quality changes, including increased wet gluten content. Increasing nitrogen fertilisation with $N_2$ resulted in an increase in the wet gluten content in the hybrid cultivars in the range from 10.2 (Hyvento) to 15.6% (Hypocamp). In the population cultivars, the increase ranged from 7.9 (Hondia) to 11.4% (Belissa). In studies by Jaskulska et al. [8], increasing nitrogen fertilisation from 100 to 200 kg/ha resulted in a significant increase in gluten in the flour by 17.6%, while Wojtkowiak et al. [39] showed no effect of the cultivar genotype and nitrogen dose on the content of wet gluten for the tested wheat cultivars. The Hyfi and Hyvento hybrid cultivar were characterised by a significantly higher ($N_1$, $N_2$) value of wet gluten compared to the population cultivars and other hybrid cultivars. The lowest ($N_1$) wet gluten content among all cultivars was found in the flour of Hyking and Hybery hybrid cultivars. A similar trend was observed for the differences between the tested cultivars fertilised with $N_1$ and $N_2$ doses in terms of dry gluten, which ranged from 7.9 to 11.0 [40]. The range of wet and dry gluten content in the flour of the studied hybrid wheat cultivars was similar to the values of these parameters reported by Taner et al. [41] and Jaskulska et al. [8]. However, higher contents of wet gluten in the range from 28.3 to 37.0% were found by Šip et al. [42]. The gluten index is the parameter used to determine the quality of washed gluten. According to Šekularac et al. [26], the cultivar genotype is the main factor for the statistically significant variability of this parameter. In the conducted study, no significant influence of cultivar (with the exception for Hybery and Hyventohybrid cultivars) and N fertilisation (with the exception for the Hyvento hybrid cultivar and the Hondiapopulation cultivar) on the values of this parameter could be determined. According to Ćurić et al. [5] GI values between 75 and 90 ensure the optimum bread baking quality. The GI value in the flour of the majority of hybrid cultivars ranged from 94 to 99, indicating too strong flour to be applicable for baking. The flour of the Hyvento hybrid cultivar was assessed markedly more favourably, as its GI remained in the optimum range from 83 ($N_1$) to 92 ($N_2$). The study results show a negative linear correlation between GI and bread specific volume (r = −0.64, $p < 0.05$), agreeing with the results obtained by Šekularac et al. [26]. Compared to the population cultivars, the Hyfi hybrid cultivar ($N_1$, $N_2$) was characterized by a higher

total ash content. The Hyfi and Hyvento hybrid cultivars ($N_1$, $N_2$) compared to the population cultivars were characterised by a significantly higher value of wet gluten content. The flour of the Hyvento hybrid cultivar was definitely more advantageous as compared to the population cultivars as the GI was in the optimal range from 83 ($N_1$) to 92 ($N_2$).

**Table 2.** Quality parameters of wheat flour from hybrid and population cultivars.

| Cultivar | Nitrogen | Content in Flour [%] | | | Gluten Index [%] | Falling Number [s] |
|---|---|---|---|---|---|---|
| | | Total Ash | Wet Gluten | Dry Gluten | | |
| Hybery | $N_1$ | 0.45 [a–d] ± 0.00 | 24.1 [a,b] ± 1.4 | 8.4 [a,b] ± 0.4 | 99 [e] ± 0 | 314 [d] ± 4 |
| | $N_2$ | 0.40 [a] ± 0.07 | 27.4 [d–g] ± 0.1 | 9.3 [d–g] ± 0.4 | 98 [c–e] ± 1 | 296 [c] ± 6 |
| Hyena | $N_1$ | 0.52 [c–e] ± 0.04 | 25.7 [b–e] ± 2.5 | 8.7 [b–d] ± 0.6 | 96 [c–e] ± 2 | 389 [h] ± 5 |
| | $N_2$ | 0.44 [a–c] ± 0.02 | 28.6 [f–h] ± 0.6 | 9.7 [f–h] ± 0.1 | 96 [b–e] ± 0 | 387 [g,h] ± 3 |
| Hyfi | $N_1$ | 0.59 [e] ± 0.02 | 27.1 [d–g] ± 0.1 | 9.2 [c–g] ± 0.2 | 96 [c–e] ± 2 | 210 [b] ± 0 |
| | $N_2$ | 0.73 [f] ± 0.03 | 30.3 [h,i] ± 0.5 | 10.2 [h,i] ± 0.2 | 94 [b,c] ± 1 | 190 [a] ± 1 |
| Hyking | $N_1$ | 0.69 [f] ±0.02 | 22.2 [a] ± 0.1 | 7.9 [a] ± 0.1 | 99 [e] ± 0 | 309 [d] ± 2 |
| | $N_2$ | 0.58 [e] ± 0.04 | 25.2 [b–d] ± 0.4 | 9.0 [b–f] ± 0.3 | 99 [e] ± 0 | 310 [d] ± 1 |
| Hymalaya | $N_1$ | 0.46 [a–d] ± 0.02 | 24.6 [b,c] ± 1.2 | 8.5 [a–c] ± 0.3 | 98 [d,e] ± 1 | 346 [f] ± 3 |
| | $N_2$ | 0.54 [d,e] ± 0.09 | 28.2 [f–h] ± 0.8 | 9.7 [f–h] ± 0.2 | 96 [c–e] ± 1 | 324 [e] ± 2 |
| Hypocamp | $N_1$ | 0.43 [a,b] ± 0.04 | 24.4 [a–c] ± 0.4 | 8.3 [a,b] ± 0.0 | 97 [c–e] ± 2 | 291 [c] ± 3 |
| | $N_2$ | 0.41 [a,b] ± 0.02 | 28.2 [f–h] ± 0.5 | 9.6 [f–h] ± 0.0 | 96 [c–e] ± 0 | 292 [c] ± 1 |
| Hyvento | $N_1$ | 0.46 [a–d] ± 0.05 | 29.4 [g,h] ± 1.2 | 9.8 [g,h] ± 0.2 | 83 [a] ± 3 | 378 [g] ± 2 |
| | $N_2$ | 0.46 [a–d] ± 0.05 | 32.4 [i] ± 0.4 | 11.0 [j] ± 0.4 | 92 [b] ± 3 | 384 [g,h] ± 4 |
| Belissa [1] | $N_1$ | 0.53 [c–e] ± 0.04 | 25.7 [c–f] ± 1.6 | 8.9 [b–e] ± 0.4 | 95 [b–d] ± 2 | 385 [g,h] ± 1 |
| | $N_2$ | 0.50 [b–e] ± 0.00 | 29.0 [I] ± 1.2 | 9.5 [i,j] ± 0.5 | 97 [c–e] ± 1 | 399 [i] ± 3 |
| Hondia [1] | $N_1$ | 0.46 [a–d] ± 0.02 | 26.8 [e–g] ± 0.8 | 9.4 [e–g] ± 0.0 | 96 [c–e] ± 1 | 403 [i] ± 3 |
| | $N_2$ | 0.58 [e] ± 0.03 | 29.1 [h,i] ± 0.8 | 9.2 [h,i] ± 0.1 | 93 [b] ± 1 | 384 [g,h] ± 4 |
| CV (%) ** | $N_1$ | 16.41 | 8.89 | 7.33 | 5.17 | 17.92 |
| | $N_2$ | 20.53 | 8.48 | 6.31 | 2.39 | 19.98 |

[1] Population cultivars. The results are presented as mean values ± standard deviation. Different letters in the same column indicate significant differences ($p = 0.05$), according to ANOVA followed by Duncan test. ** CV coefficient of variation; $N_1$—110 kg/ha. $N_2$—150 kg/ha.

The values of the coefficient of variation (CV) for the $N_1$ and $N_2$ doses were similar for the wet gluten as well as the dry gluten and falling number. The use of a higher dose of $N_2$ increased the stability of the gluten index whereas the ash content was characterised by higher variability.

### 3.3. Flour Water Absorption and Rheological Properties of Dough

Determination of flour water absorption, as well as the results of the faringraphic analysis of the flour play a key role in the assessment of wheat flour baking quality [36,43]. The results of determinations of these parameters are presented in Table 3.

Water absorption of the flour of the hybrid cultivars was considerably lower than for the Belissa population cultivar. However, the Hymalaya, Hyena, Hyfi, and Hyking hybrid cultivars were found to have similar or higher values of this parameter than the Hondia population cultivar. The influence of the cultivar factor on significant differentiation of the flour water absorption and other traits of the wheat dough was also confirmed by Warechowska et al. [29] and Silva et al. [44]. Increased water absorption of flour with the increased N dose was determined at the range between 0.9 and 2.0% in the population cultivars and between 1.5 and 2.1% in the hybrid cultivars. Only for the Hyfi hybrid cultivar, the increased water absorption of flour was 0.6% for $N_2$ compared to $N_1$. Park et al. [7] also recorded increased wheat water absorption of flour under the influence of increased nitrogen fertilisation, but such a relationship was not shown by Warechowska et al. [29]. A negative linear correlation was observed between water absorption of flour and specific volume of its bread (r = −0.64, $p < 0.05$) and positive between water absorption of flour and dough yield (r = 0.62, $p < 0.05$).

**Table 3.** Results of rheological analysis of wheat flour obtained from hybrid and population cultivars.

| Cultivar | Nitrogen | Water Absorption of Flour [%] | Properties of Dough | | | |
|---|---|---|---|---|---|---|
| | | | Development Time [min] | Stability [min] | Degree of Softening [FU] | Quality Number |
| Hybery | $N_1$ | 54.0 [a] ± 0.3 | 2.1 [a–c] ± 0.1 | 4.5 [e,f] ± 1.3 | 74 [a,b] ± 4 | 70 [e–g] ± 4 |
| | $N_2$ | 56.0 [c] ± 0.1 | 2.7 [a–c] ± 0.5 | 5.6 [f,g] ± 0.1 | 78 [a] ± 1 | 73 [i–j] ± 4 |
| Hyena | $N_1$ | 57.5 [d,e] ± 0.1 | 2.5 [a–c] ± 0.3 | 4.1 [b–f] ± 0.1 | 97 [a,b] ± 4 | 56 [e–g] ± 1 |
| | $N_2$ | 59.0 [g] ± 0.0 | 2.6 [a–c] ± 0.6 | 3.9 [a–e] ± 0.8 | 102 [b,c] ± 0 | 52 [c–e] ± 4 |
| Hyfi | $N_1$ | 57.9 [e] ± 0.1 | 3.0 [a–c] ± 0.4 | 2.9 [a–d] ± 0.3 | 125 [e] ± 2 | 47 [a–d] ± 2 |
| | $N_2$ | 58.5 [f] ± 0.2 | 3.3 [a–c] ± 0.6 | 2.7 [a,b] ± 0.2 | 129 [d,e] ± 11 | 46 [a–d] ± 4 |
| Hyking | $N_1$ | 57.5 [d,e] ± 0.1 | 2.4 [a–c] ± 0.5 | 5.3 [e–g] ± 0.4 | 78 [a] ± 7 | 72 [i–j] ± 2 |
| | $N_2$ | 58.5 [f] ± 0.0 | 2.9 [a–c] ± 0.5 | 6.6 [g] ± 0.7 | 75 [a] ± 4 | 76 [j] ± 1 |
| Hymalaya | $N_1$ | 56.0 [c] ± 0.1 | 2.5 [a–c] ± 0.4 | 5.1 [e,f] ± 0.8 | 86 [a,b] ± 6 | 62 [b–f] ± 3 |
| | $N_2$ | 57.8 [e] ± 0.1 | 3.9 [c] ± 0.7 | 5.2 [e–g] ± 0.1 | 93 [a,b] ± 2 | 68 [h–l] ± 3 |
| Hypocamp | $N_1$ | 55.1 [b] ± 0.1 | 2.1 [a–c] ± 0.1 | 2.9 [a–d] ± 0.0 | 120 [c–e] ± 1 | 44 [a–c] ± 1 |
| | $N_2$ | 57.2 [d] ± 0.4 | 1.9 [a] ± 0.3 | 2.5 [a] ± 0.2 | 128 [e] ± 3 | 43 [a,b] ± 1 |
| Hyvento | $N_1$ | 54.8 [b] ± 0.2 | 2.9 [a–c] ± 0.3 | 5.1 [e,f] ± 0.4 | 87 [a,b] ± 1 | 63 [g,h] ± 3 |
| | $N_2$ | 56.4 [c] ± 0.1 | 3.6 [b,c] ± 0.1 | 4.3 [d–f] ± 0.1 | 89 [a,b] ± 1 | 61 [f–h] ± 0 |
| Belissa [1] | $N_1$ | 59.0 [g] ± 0.1 | 2.1 [a,b] ± 0.5 | 2.8 [a–c] ± 1.1 | 135 [e] ± 3 | 39 [a] ± 7 |
| | $N_2$ | 61.0 [h] ± 0.4 | 2.9 [a–c] ± 0.5 | 2.8 [a–d] ± 0.4 | 134 [e] ± 2 | 45 [a–c] ± 8 |
| Hondia [1] | $N_1$ | 56.2 [c] ± 0.1 | 1.8 [a] ± 0.4 | 4.3 [d–f] ± 0.1 | 95 [a,b] ± 4 | 54 [d–f] ± 1 |
| | $N_2$ | 57.1 [d] ± 0.1 | 2.1 [a,b] ± 0.4 | 4.1 [b–f] ± 0.4 | 105 [b–d] ± 4 | 50 [b–e] ± 6 |
| CV (%) ** | $N_1$ | 2.82 | 23.76 | 27.98 | 22.80 | 20.12 |
| | $N_2$ | 2.55 | 32.47 | 34.09 | 21.75 | 22.34 |

[1] Population cultivars. The results are presented as mean values ± standard deviation. Different letters in the same column indicate significant differences ($p = 0.05$), according to ANOVA followed by Duncan test. ** CV coefficient of variation; $N_1$—110 kg/ha. $N_2$—150 kg/ha.

The dough development time remained in the range from 1.8 to 3.9 min. Significantly longer dough development time characterised only the Hymalaya and Hyvento hybrid cultivars, whereas the lowest value of this parameter was determined in the flour of the Hondia population cultivar. Low dough stability from 2.5 to 2.9 min characterised the Hyfi and Hypocamp hybrid cultivars, as well as the Belissa population cultivar. The character of the dough was considerably more favourably assessed in the tested Hybery, Hyking and Hymalaya hybrid cultivars. Dough stability in these cultivars was at least one minute or twofold longer than for the Hondia and Belissa population cultivars, which corresponds to strong flour of the three mentioned hybrid wheat cultivars. It was noted that the dough stability parameter of the Hyking cultivar ($N_1$) was significantly higher than the value of this parameter in the tests of the Hyfi, Hypocamp and Belissa dough, made from flour obtained from the experiment with a higher level of nitrogen fertilisation ($N_2$). [40,45]. As stated by Silva et al. [44], extension of the dough development time and stability is caused by increased total protein content, and dough with such parameters is obtained from flour with high gluten and HMW glutenin content. In the present study, a positive linear correlation was observed between dough development time and crude protein content (r = 0.63, $p < 0.05$), wet gluten (r = 0.65, $p < 0.05$) and dry gluten (r = 0.60, $p < 0.05$). In terms of dough softening, the lowest score was assigned to the Belissa population cultivar and the Hyfi and Hypocamphybrid cultivars, whereas the value of this parameter for these cultivars was 130 FU, on average. Dough mixing qualities are considered satisfactory when the degree of softening is below 70 FU [27]. The Hybery and Hyking hybrid cultivars were characterised by a degree of softening favourable in terms of dough quality, which was lower by 50 FU. Dough from these cultivars was characterised by better stability and a lower degree of softening, with a higher quality number. The significant increase of dough stability and degree of softening under the impact of the increased N dose known from the study of Park et al. [7] and Ma et al. [10] could not be confirmed. The Hyfi, Hypocamp, and Belissa cultivars fertilised with $N_1$ had a higher degree of dough softening than Hybery, Hyking, Hymalaya and Hyvento fertilised with $N_2$. The flour of the Hyking cultivar, both in

the $N_1$ and $N_2$ variants, was distinguished by the highest quality number. The grain of the Hyena, Hyfi, Hypocamp, Hyvento, Belissa, and Hondia cultivars obtained under conditions of higher nitrogen fertilisation ($N_2$) did not match the quality number of the Hyking cultivar ($N_1$), which indicates that the cultivar's factor allows the studied cultivars to be distinguished in terms of their potential suitability for food processing. Statistical analysis did not confirm significant differentiation of the remaining rheological traits of the dough, for hybrid as well as population cultivars with an increased N dose, which, in contrast, coincides with the results obtained by Silva et al. [44]. Strong negative linear correlations were observed between the quality number and degree of softening, as well as between the dough stability and degree of softening [40]. Both values of the correlation coefficient were r = −0.94, *p* < 0.05. Moreover, a strong positive linear correlation between quality number and dough stability was demonstrated (r = 0.96, *p* < 0.05). The use of the $N_2$ dose resulted in an increase in the dough development time in both hybrid (except the Hypocampcultivar) and population cultivars. The $N_2$ dose influenced the increase of dough stability only in the Hybery, Hyking and Hymalayahybrid cultivars.

The dough quality parameters were characterised by high volatility, except for the coefficient of variation (CV) for water absorption ($N_1$ = 2.82%, $N_2$ = 2.55%). The increase in nitrogen dose from 110 to 150 kg/ha resulted in greater differentials in the dough development time, dough softening, and quality number.

### 3.4. The Flour Baking Quality Assessment

The flour baking quality assessment was completed via the laboratory baking method, comparing the results of the course of individual dough production phases and the assessment of the obtained product [5,6]. The dough from Belissa population cultivar was characterised by a significantly higher dough yield, and the Hybery hybrid cultivar had the lowest compared to the remaining cultivars (Table 4). The difference in terms of this parameter in the assessment of the baking process of the dough from Belissa population cultivar and Hybery hybrid cultivar flour was 13.0%. In the conducted study, as in the study by Jaskulska et al. [8] on winter wheat, a positive impact of the $N_2$ dose on dough yield occurred. The total baking loss was similar in the majority of cultivars, and for the hybrid cultivars, the lowest value was observed for the Hyfi cultivar ($N_1$) and highest for the Hyvento cultivar ($N_2$). The dough yield obtained from the hybrid flour ranged between 142.3 and 148.5 $cm^3$. The lower the baking loss, the higher the bread yield that was observed for hybrid the Hybery ($N_1$, $N_2$), Hyfi ($N_2$) and Hymalaya cultivars ($N_2$), without an impact of the N doses on the value of this parameter. The bread yield and Dallmann porosity index of the crumb were positively correlated (r = 0.48 and r = 0.49, *p* < 0.05) with the gluten index. Bread volume is one of the basic traits taken into account for bread quality assessment and it was slightly variable for the hybrid wheat cultivars. The Hybery and Hyvento hybrid cultivars had significantly higher value of this parameter than the remaining cultivars. The Hyvento hybrid cultivar was characterised by the lowest bread yield and the concomitant highest gluten quality (GI < 95), which was confirmed by Ćurić et al. [5] who indicated that the GI value is strongly correlated to bread quality. The influence of the cultivar's factor was also noted in the comparison of the bread yield parameter from the flour of the Hypocamp hybrid cultivar obtained from grains cultivated with a higher nitrogen dose ($N_2$), it was much lower than that obtained with the flour of the Hybery hybrid cultivar cultivated with a lower level of nitrogen fertilisation ($N_1$). A significant impact of N dose on bread volume was determined only for the Hymalaya and Hyvento hybrid cultivars. However, a lower bread volume, by 12.5 and 13.5% respectively, characterised the breads made from the flour of these cultivars fertilised with the $N_2$ dose. Jaskulska et al. [8] also were unable to demonstrate a direct effect of N fertilisation on bread yield and volume, whereas the same effect was confirmed by Grahmann et al. [6]. The Dallmann porosity index of crumb was good or very good. Also, in the case of the Hybery and Hyvento hybrid cultivars, the crumb of breads with the highest volume received the best score. The amount of crumb moisture content depends on a variety of factors, including the recipe and baking parameters, which is confirmed in the presented results, as the

mentioned cultivars were distinguished by higher water absorption of the flour, and this parameter determines the amount of water in the recipe for the bread dough. The crumb moisture content from wheat flour obtained from the grain ($N_1$) of the hybrid cultivars was undifferentiated, and a higher crumb moisture content was found for Belissa and Hondia population cultivars. However, in the case of the $N_2$ dose, a higher crumb moisture content was observed in the breads made from the flour from the Hybery, Hyena, Hymalaya, and Hypocamp hybrid cultivars. Statistical analysis revealed a positive linear correlation between crumb moisture content and dough yield ($r = 0.65$, $p < 0.05$). The lack of a significant impact of the $N_2$ dose as compared to the $N_1$ on the assessment of bread volume, the Dallmann porosity index of crumb, and the crumb moisture content from hybrid cultivars suggest the possibility of obtaining flour of very good baking value with a lower N dose used [4]. A study by Vazquez et al. [43] showed that it is possible to modulate the wheat grain and flour for bread according to the nitrogen regime, although it is necessary to understand the genotype-environment relationship. Increasing the content of protein in the grain under the impact of nitrogen fertilisation does not directly affect the increased volume of the bread made from flour obtained from such grain [4]. It is important to minimise the environmental issues associated with late N usage, thus it is significant to know what dosage and fertiliser form, as well as date of N fertilisation would be necessary to obtain the desired baking traits of wheat cultivars [43]. Among the assessed characteristics of the baking process, the application of the $N_2$ dose increased only in the case of the dough yield.

The assessed quality indicators for bread were characterised by high stability and the values of the coefficient of variation (CV) were the lowest for bread yield and crumb moisture content. The use of the $N_1$ dose increased the stability of baking loss while the $N_2$ dose increased the stability of bread volume and the Dallmann porosity index of crumb. Photographs of the bread crumb of selected cultivars are shown in Figure 1.

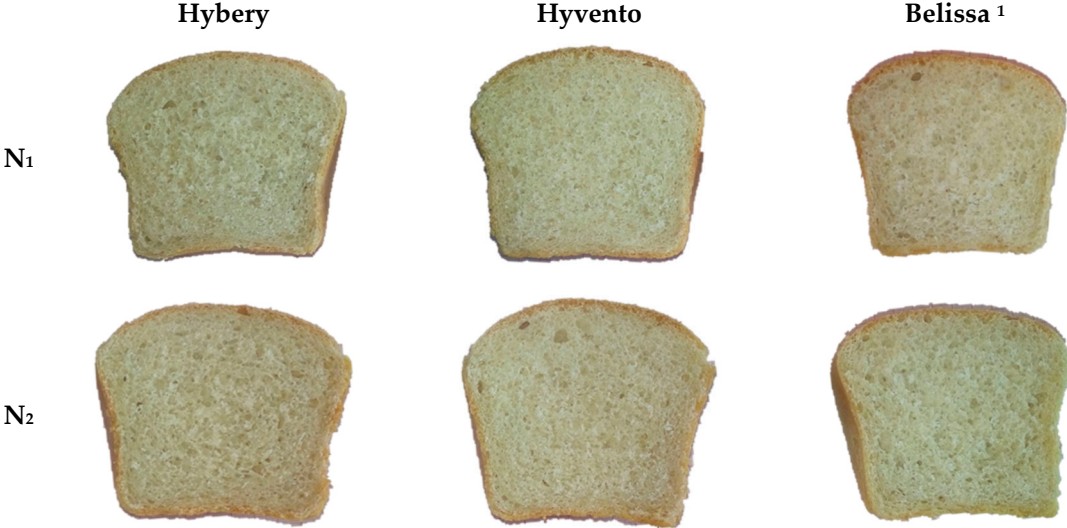

**Figure 1.** Crumb porosity of bread obtained from the flours of selected wheat cultivars with various levels of fertilisatio: $N_1$.—110 kg/ha, $N_2$—150 kg/ha. [1] Population cultivars.

**Table 4.** Baking process parameters of wheat flour and quality indicators of wheat bread obtained from hybrid and population cultivars.

| Cultivar | Nitrogen | Dough Yield [%] | Baking Loss [%] | Bread Yield [%] | Bread Volume [cm$^3$/g] | Dallmann Porosity Index of Crumb [Scores] | Crumb Moisture Content [%] |
|---|---|---|---|---|---|---|---|
| Hybery | $N_1$ | 143.8 $^a$ ± 0.3 | 15.1 $^{a,b}$ ± 0.9 | 148.1 $^{c,d}$ ± 1.6 | 3.2 $^{c-e}$ ± 0.2 | 100 $^c$ ± 0 | 43.9 $^{b-d}$ ± 0.4 |
|  | $N_2$ | 145.5 $^b$ ± 0.1 | 15.1 $^{a,b}$ ± 0.1 | 148.1 $^{c,d}$ ± 0.1 | 3.3 $^e$ ± 0.1 | 100 $^c$ ± 0 | 45.0 $^{e-g}$ ± 0.4 |
| Hyena | $N_1$ | 148.6 $^{e,f}$ ± 0.0 | 15.9 $^{a-c}$ ± 0.5 | 146.5 $^{b-d}$ ± 0.8 | 2.9 $^{a,b}$ ± 0.0 | 95 $^{b,c}$ ± 7 | 43.1 $^{a,b}$ ± 0.6 |
|  | $N_2$ | 149.3 $^{f,g}$ ±0.5 | 16.2 $^{a-d}$ ± 0.9 | 146.2 $^{a-d}$ ± 1.6 | 2.9 $^{a,b}$ ± 0.0 | 90 $^{a-c}$ ± 0 | 44.3 $^{c-f}$ ± 0.3 |
| Hyfi | $N_1$ | 146.2 $^{f,g}$ ±0.4 | 17.1 $^{a-d}$ ± 0.5 | 144.5 $^{a-d}$ ± 0.9 | 2.9 $^{a,b}$ ± 0.0 | 90 $^{a-c}$ ± 0 | 43.6 $^{a-c}$ ± 0.3 |
|  | $N_2$ | 149.6 $^{b-d}$ ± 1.2 | 14.8 $^a$ ± 0.5 | 148.5 $^d$ ± 1.2 | 3.0 $^{a-d}$ ± 0.1 | 90 $^{a-c}$ ± 0 | 44.2 $^{c-e}$ ± 0.6 |
| Hyking | $N_1$ | 148.9 $^{f,g}$ ± 0.4 | 16.0 $^{a-d}$ ± 0.4 | 146.4 $^{a-d}$ ± 0.6 | 3.0 $^{a-d}$ ± 0.0 | 85 $^{a,b}$ ± 7 | 43.9 $^{b-d}$ ± 0.7 |
|  | $N_2$ | 150.0 $^g$ ± 0.4 | 16.4 $^{a-d}$ ± 0.7 | 145.8 $^{a-d}$ ± 1.3 | 3.0 $^{a-c}$ ± 0.2 | 90 $^{a-c}$ ± 0 | 44.5 $^{d-g}$ ± 0.2 |
| Hymalaya | $N_1$ | 146.0 $^{b,c}$ ± 0.4 | 17.0 $^{a-d}$ ± 0.9 | 144.6 $^{a-d}$ ± 1.5 | 3.2 $^{d,e}$ ± 0.0 | 85 $^{a,b}$ ± 7 | 43.2 $^{a,b}$ ± 0.2 |
|  | $N_2$ | 147.0 $^{c,d}$ ± 0.2 | 15.3 $^{a,b}$ ± 0.2 | 147.7 $^{c,d}$ ± 0.4 | 2.8 $^a$ ± 0.0 | 90 $^{a-c}$ ± 0 | 44.1 $^{c,d}$ ± 0.5 |
| Hypocamp | $N_1$ | 147.7 $^{d,e}$ ± 0.2 | 17.2 $^{b-d}$ ± 0.6 | 144.3 $^{a-c}$ ± 1.1 | 3.1 $^{b-d}$ ± 0.1 | 95 $^{b,c}$ ± 7 | 43.2 $^{a,b}$ ± 0.1 |
|  | $N_2$ | 154.3 $^h$ ± 0.5 | 17.8 $^{c,d}$ ± 0.1 | 143.5 $^{a,b}$ ± 0.1 | 3.1 $^{b-d}$ ± 0.1 | 95 $^{b,c}$ ± 7 | 44.7 $^{d-g}$ ± 0.2 |
| Hyvento | $N_1$ | 145.6 $^b$ ± 0.1 | 18.4 $^d$ ± 0.2 | 142.3 $^a$ ± 0.4 | 3.7 $^f$ ± 0.1 | 96 $^a$ ± 0 | 43.1 $^{a,b}$ ± 0.3 |
|  | $N_2$ | 146.2 $^{b,c}$ ± 0.1 | 16.8 $^{a-d}$ ± 0.2 | 142.9 $^{a-d}$ ± 0.6 | 3.2 $^{c-e}$ ± 0.1 | 98 $^{a,b}$ ± 7 | 43.0 $^a$ ± 0.1 |
| Belissa [1] | $N_1$ | 156.7 $^i$ ± 0.9 | 16.8 $^{a-d}$ ± 0.2 | 145.1 $^{a-d}$ ± 0.3 | 2.9 $^{a,b}$ ± 0.1 | 95 $^{b,c}$ ± 7 | 45.2 $^g$ ± 0.2 |
|  | $N_2$ | 158.3 $^j$ ± 0.7 | 16.7 $^{a-d}$ ± 0.1 | 145.3 $^{a-d}$ ± 0.2 | 3.0 $^{a-d}$ ± 0.0 | 93 $^{b,c}$ ± 4 | 45.1 $^{f,g}$ ± 0.1 |
| Hondia [1] | $N_1$ | 154.5 $^h$ ± 0.6 | 16.7 $^{a-d}$ ± 0.2 | 145.4 $^{a-d}$ ± 0.4 | 3.0 $^{a-d}$ ± 0.0 | 95 $^{b,c}$ ± 7 | 44.3 $^{c-f}$ ±0.1 |
|  | $N_2$ | 155.8 $^i$ ± 0.5 | 16.6 $^{a-d}$ ± 0.1 | 145.5 $^{a-d}$ ± 0.1 | 3.1 $^{b-d}$ ± 0.1 | 95 $^{b,c}$ ± 7 | 44.9 $^{e-g}$ ± 0.1 |
| CV (%) ** | $N_1$ | 2.78 | 5.95 | 1.19 | 8.29 | 7.01 | 1.65 |
|  | $N_2$ | 3.05 | 7.98 | 1.69 | 5.12 | 4.85 | 1.47 |

[1] Population cultivars. The results are presented as mean values ± standard deviation. Different letters in the same column indicate significant differences ($p = 0.05$), according to ANOVA followed by Duncan test. ** CV coefficient of variation; $N_1$—110 kg/ha. $N_2$—150 kg/ha.

### 3.5. Principal Component Analysis

The relationship between the quality parameters measured in nine winter wheat cultivars fertilised with variable nitrogen dosage has been assessed by e principal component analysis (PCA) in order to estimate the source of variability. Among the twenty-one tested parameters, for the purpose of analysis eight extracted principal components were selected having eigenvalues higher than the average. The eigenvalues for the first component was 3.73, and the percentage of the explained variance was 46.56%. The second component explains 26.12% of the variance, and its own value was 2.09 (Table 5). The first two components transfer 72.69% of the variability of the original data. In order to determine the number of main components, the criterion of eigenvalue greater than the unity of the so-called Kaiser criterion was used [46]. It is therefore possible to approximate the original data set only in two dimensions. A scree plot confirms the significance of the first two components, which allows the reduction of the 8-dimensional space to two components (Figure 2).

**Table 5.** Individual values of correlation matrix.

| Value Number | Eigenvalues (Correlations), Related Statistics | | | |
|:---:|:---:|:---:|:---:|:---:|
| | Eigenvalue | % of Total Variance | Cumulative Eigenvalue | Cumulative% |
| 1 | **3.725399** | **46.56749** | **3.725399** | **46.5675** |
| 2 | **2.089800** | **26.12250** | **5.815199** | **72.6900** |
| 3 | 0.792959 | 9.91199 | 6.608158 | 82.6020 |
| 4 | 0.618540 | 7.73174 | 7.226698 | 90.3337 |
| 5 | 0.439531 | 5.49414 | 7.666229 | 95.8279 |
| 6 | 0.250196 | 3.12745 | 7.916425 | 98.9553 |
| 7 | 0.062311 | 0.77889 | 7.978736 | 99.7342 |
| 8 | 0.021264 | 0.26580 | 8.000000 | 100.0000 |

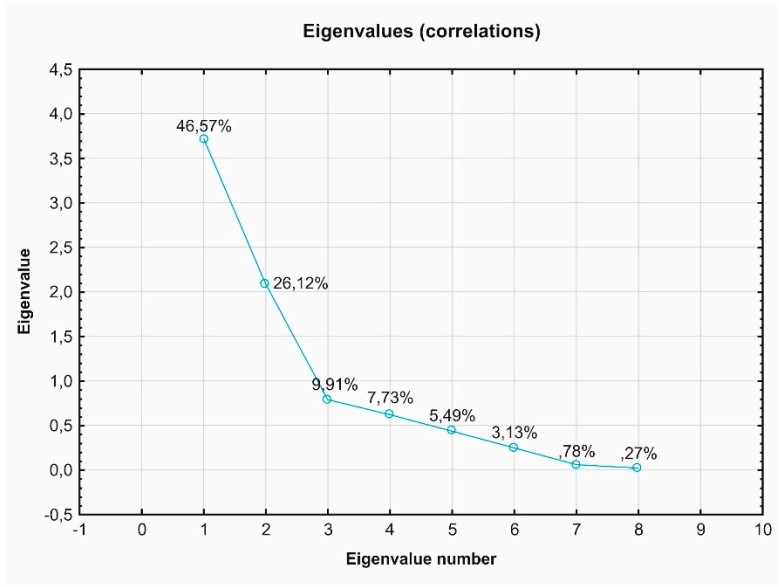

**Figure 2.** Scree plot for individual values of the matrix of correlation.

Figure 3a shows the projection of variables on the factor plane. The first principal component transfers the information contained in positively correlated variables, degree of softening, dough yield and negatively correlated values, dough stability, quality number and grain vitreousness (Table 6). The variable quality number is positively correlated with dough stability ($r = 0.96$, $p < 0.05$), while the variable dough yield is positively correlated with the variable dough degree of softening ($r = 0.58$, $p < 0.05$), which indicates the position of the vectors in close proximity. The second principal component comprises mostly of positively correlated variables, Dallmann porosity index of crumb, gluten index

and bread yield. The variables of bread yield and Dallmann porosity index of crumb are positively correlated with the gluten index variable (r = 0.48 and r = 0.49, *p* < 0.05). The vectors of the degree of softening, dough quality number and grain vitreousness variables are longer and located closer to the circle, which indicates that these variables contribute the most information.

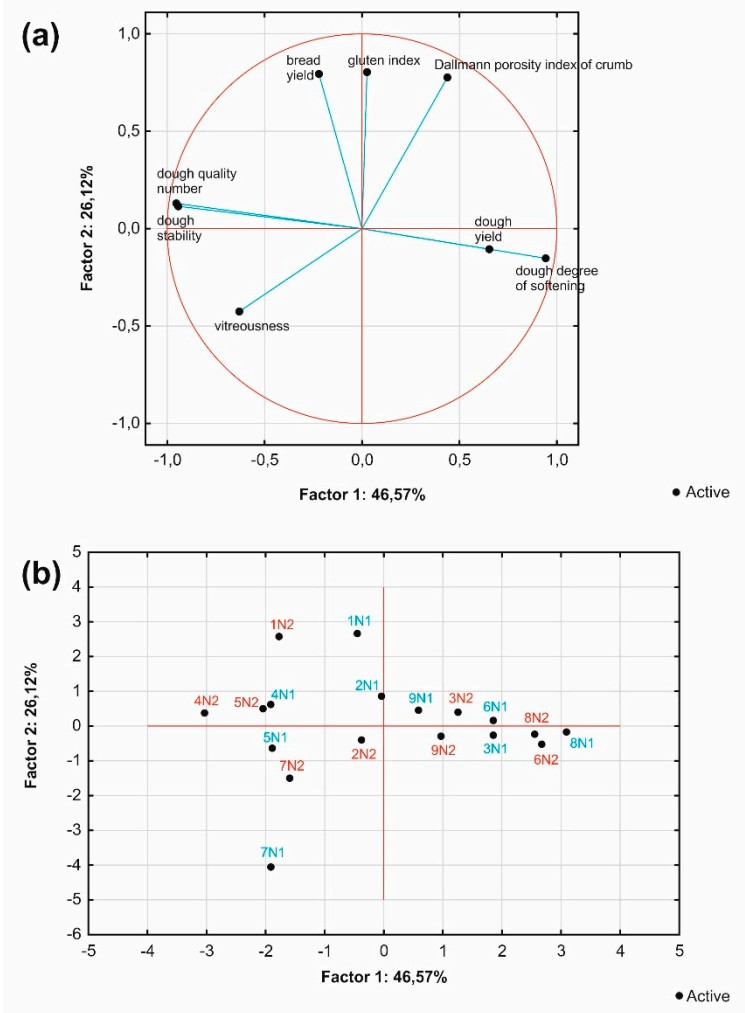

**Figure 3.** Principal components analysis of (**a**) distribution of the analyzed parameters (**b**) distribution of wheat cultivars and nitrogen fertilisation. 1—Hybery. 2—Hyena. 3—Hyfi. 4—Hyking. 5—Hymalaya. 6—Hypocamp. 7—Hyvento. 8—Belissa. 9—Hondia. N1—110 kg/ha. N2—150 kg/ha.

**Table 6.** Value of principal components coefficients.

| Variable | Factor Coordinates of Variables, Based on Correlation | |
| --- | --- | --- |
| | **Factor 1** | **Factor 2** |
| dough degree of softening | 0.935932 | −0.147075 |
| dough stability | 0.651097 | −0.107195 |
| Dallmann porosity index of crumb | 0.433313 | 0.771696 |
| gluten index | 0.024504 | 0.797857 |
| bread yield | −0.222052 | 0.788339 |
| dough stability | −0.940939 | 0.117195 |
| quality number | −0.951258 | 0.129446 |
| vitreousness | −0.630539 | −0.415473 |

Figure 3b presents the project of the analysed winter wheat cultivars fertilised with different nitrogen doses ($N_1$ and $N_2$) on the factor plane. Of the studied hybrid cultivars, the Hybery cultivar ($N_1$ and $N_2$) had the best applicability for baking process, whereas the Hyvento cultivar ($N_1$) was characterised by favourable grain quality properties, dough rheological properties and bread volume, but its remaining baking quality parameters were less optimal. The proximity of the remaining cultivars confirms their similarity in terms of the analysed quality parameters.

## 4. Conclusions

Grain of the studied wheat cultivars was characterised by a variable technological property. Grain of the Hypocamp hybrid cultivar had a more favourable assessment in terms of grain bulk density ($N_1$ and $N_2$) and flour yield ($N_2$), the Hymalaya cultivar in terms of vitreousness ($N_1$ and $N_2$) and flour yield ($N_1$), whereas the Hyvento cultivar in terms of vitreousness ($N_1$ and $N_2$) and crude protein content ($N_2$). Increase of fertilisation to $N_2$ resulted in a significant increase of crude protein content in the case of all of the studied cultivars. Significant variability of falling number and water absorption of the flour was observed between cultivars fertilised with different nitrogen doses. The nitrogen dose increased to $N_2$ resulted in increased wet and dry gluten content, and the highest value of these parameters, similar to the population cultivars, was found in the Hyfi and Hyvento hybrid cultivars. The flour obtained from the Belissa population cultivar grain was distinguished by higher water absorption than in the remaining cultivars, whereas the Hyking hybrid cultivar flour had the highest quality number. The least favourable assessment of the rheological properties concerned the dough of the Hyfi and Hypocamp hybrid cultivars and the Belissa population cultivar.

The hybrid wheat flour was characterised by good quality, therefore it can be used for wheat bread production. No influence of the increased $N_2$ dose on the baking process and bread quality could be found with the exception of crumb moisture content and dough yield. Breads obtained from Hybery and Hyvento hybrid cultivars were characterised by greater volume than Belissa and Hondia population cultivars.

The greatest applicability for baking purposes was determined to be Hybery hybrid cultivar due to high values of the following parameters: -bread yield, specific bread volume, and low baking loss, which may contribute to increased interest in this cultivar by bread producers. Without a doubt, this may result in increased purchasing prices of hybrid cultivar grain, which ultimately justifies the purchase of more expensive grain of hybrid wheat cultivars. The research on hybrid wheat cultivars grown in different climatic and agronomic conditions have produced promising results, and considering the shrinking cultivation areas and climate change, such as prolonged droughts, it seems necessary to maintain the leading role of cereals in global plant production.

**Author Contributions:** Conceptualization, M.J.-P.; J.B. and J.K.; methodology, M.J.-P.; J.B. and J.K.; investigation, M.J.-P.; J.K.; writing—original draft preparation, M.J.-P.; J.K.; and J.B.; writing—review and editing, M.J.-P.; J.B.; J.K.; E.S.-K.; D.B.-J. and G.J. All authors have read and agreed to the published version of the manuscript.

**Funding:** The project was supported by the Minister of Science and Higher Education of Poland (Project No.026/RID/2018/19 "Regional Initiative of Excellence", 2019–2022).

**Conflicts of Interest:** The authors declare no conflict of interest.

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
