# Peer review of "A Comparative Assessment of the Baking Quality of Hybrid and Population Wheat Cultivars"

_applsci, doi:10.3390/app10207104_

Round 1

Reviewer 1 Report

The manuscript is well presented, and the results are interesting. Some minor points should be addressed.

Lines 106-107: Please provide relevant humidity and temperature of the fermentation chamber

Line 108: kneading of the dough was by hand?

If applicable, please provide photographs of the bread made from different flours.

Author Response

Odpowiedź dla recenzenta 1 komentarz

Dziękuję za konstruktywne i pouczające uwagi Recenzenta. Poniżej znajdują się nasze szczegółowe odpowiedzi na wątpliwości zgłoszone podczas procesu przeglądu.

Punkt 1: Wiersze 106–107: Proszę podać odpowiednią wilgotność i temperaturę komory fermentacyjnej

Odpowiedź 1: Zmieniona zgodnie z komentarzem Recenzenta.

Punkt 2: Wiersz 108: wyrabianie ciasta było ręczne?

Odpowiedź 2: Odpowiedzi na powyższe pytanie zostały uzupełnione w rękopisie

Punkt 3: W stosownych przypadkach proszę załączyć zdjęcia chleba wykonanego z różnych mąk.

Odpowiedź 3: Zgodnie z komentarzem Recenzenta dodano zdjęcia chleba z 2 zalecanych odmian mieszańcowych (Hybery i Hyvento) oraz odmiany populacyjnej Belissa.

Reviewer 2 Report

This manuscript assessed the quality parameters of grain and flour, rheological properties of dough and quality of bread prepared from flour of hybrid cultivars of wheat in comparison with population cultivars of wheat. This manuscript revealed that Hybrid wheat flour was characterized by good quality, therefore it can be used for wheat bread production. Among the hybrid cultivars, the best applicability for baking purposes characterized Hybery cv (N1, N2) due to the favourable values of the baking process parameters and bread quality.

But this manuscript is not innovative enough. The author assessed the quality of grain and flour, baking quality and applicability for the production of bread of flour obtained from the grain of hybrid wheat cultivars. Previous studies mostly used these methods . Therefore this manuscript is not innovative enough . In the “Abstract” section ,author only described the content of the study, not the significance of the study. In the " Conclusions " section, the authors also did not elaborate on the significance of the study. I suggest that the author elaborate on the significance of the study.For example,the authors can elaborate on the specific effects of this hybrid wheat on future global plant production and research . The manuscript lacks an overview of the relevant research The text description of the manuscript is not clear enough. There are some problems inadequacies should be improved, such as:

  1. Materials and Methods:The author should explain the reason that two levels of nitrogen fertilization (N1 – 110 kg/ha, N2 – 150 kg/ha) was chosen for the study.
  2. Results and Discussion:The authors do not discuss the results in detail in this part. I suggest to discuss the research results.
  3. Line 276, I suggest to change “Laboratory Bread Baking Results” to “The flour baking quality assessment ”
  4. The authors did not elaborate on the significance of the study.
  5. The pictures in the manuscript are not clear enough.
  6. There are some grammatical errors.

7 .The text description of the manuscript is not clear enough.

8 .The manuscript lacks an overview of the relevant research.

Author Response

Response to Reviewer 2 Comments

Thank you for the constructive and informative comments of the Reviewer. Below are our point-by-point responses to the concerns raised during review process.

Point 1: This manuscript is not innovative enough. The author assessed the quality of grain and flour, baking quality and applicability for the production of bread of flour obtained from the grain of hybrid wheat cultivars. Previous studies mostly used these methods. Therefore this manuscript is not innovative enough.

Response 1: Thank you for the Reviewer's comment. The article presented for review discusses the issue of comparing the properties of grain and flour from hybrid wheat grain in comparison with population wheat grain and its products and the quality of the bread obtained from them. The area occupied by hybrid wheat accounts for less than 1.0% of the global area under wheat cultivation. In Europe, hybrid wheat is cultivated in the area of over 560,000 ha, and 80% of hybrid cultivars are grown in France, and the remaining 20% in Germany, Hungary, Italy, Czech Republic, Slovakia, Romania and Portugal. Therefore, the authors were guided by the idea of disseminating the results of research on the suitability of selected varieties for bakery production. According to the Review of the literature data, the subject is not thoroughly understood, and the obtained results of our research may contribute to this subject, because, as scientists predict, the future of wheat cultivation may be heading towards hybrid cultivars. The opinion of the Reviewer of our article is extremely valuable for us and helpful in designing further research in the field of assessing the usefulness and popularizing hybrid cultivars of wheat.

Point 2: In the “Abstract” section, author only described the content of the study, not the significance of the study.

Response 2: The “Abstract” was changed in accordance with the Reviewer’s comment.

Point 3: In the " Conclusions " section, the authors also did not elaborate on the significance of the study. I suggest that the author elaborate on the significance of the study. For example, the authors can elaborate on the specific effects of this hybrid wheat on future global plant production and research. The manuscript lacks an overview of the relevant research. The text description of the manuscript is not clear enough.

Response 3: The “Conclusions” were changed in accordance with the Reviewer’s comment.

Point 4: Materials and Methods:The author should explain the reason that two levels of nitrogen fertilization (N1 – 110 kg/ha, N2 – 150 kg/ha) was chosen for the study.

Response 4: The aim at applying diversified doses of nitrogen fertilization (N1 and N2) to the studied hybrid and population cultivars of winter wheat was to determine which cultivar containing a lower nitrogen dose (N1) had comparable or more favorable grain quality parameters to those ones with the higher nitrogen (N2) dose. These procedures made it possible to identify cultivars, especially hybrid ones, for which it is not necessary to use higher nitrogen doses. That allowed not only to reduce the production costs but also to limit the negative impact of higher nitrogen doses for the environment.

The information about fertilisation was added in manuscript – chapter 2.1. „Experimental Material”

Point 5: Results and Discussion:The authors do not discuss the results in detail in this part. I suggest to discuss the research results.

Response 5: The “Results and Discussion” were changed in accordance with the Reviewer’s comment.

Point 6: Line 276, I suggest to change “Laboratory Bread Baking Results” to “The flour baking quality assessment ”

Response 6: Changed in accordance with the Reviewer’s comment.

Point 7: The authors did not elaborate on the significance of the study.

Response 7: Changed in accordance with the Reviewer’s comment. The significance of the study were described in “Abstract” and “Conclusions”

Point 8: The pictures in the manuscript are not clear enough.

Response 8: Changed in accordance with the Reviewer’s comment.

Point 9: There are some grammatical errors.

Response 9: Changed in accordance with the Reviewer’s comment. The manuscript text was checked by a native speaker.

Point 10: The text description of the manuscript is not clear enough.

Response 10: Changed in accordance with the Reviewer’s comment.

Point 11: The manuscript lacks an overview of the relevant research.

Response 11: Changed in accordance with the Reviewer’s comment. The overview of the relevant research were added in chapter “Results and Discussion”.

Reviewer 3 Report

Title reflect the content and emphasize the paper's interest and significance.

Abstract should be supplemented by the results obtained from the conducted research (please add scientific and practical significance of the selected method). Rewrite it clearly stating the facts; focus more on how your research has contributed to knowledge gaps; describe research limitations for future research and restate your major findings.

Introduction provides a good, generalized background of the topic that quickly gives the reader an appreciation of the wide range of applications for this technology.

Materials and Methods used in this study are standard. The PCA analysis is not well written in the manuscript. The presented results are not sufficient and do not justify the application of this method in the paper. I suggest a complete revision of the section related to PCA analysis.

Results and Discussion: The results are partially explained and there are not all of them presented in an appropriate format. Please revise the results of PCA analysis.

Author Response

Response to Reviewer 3 Comments

Thank you for the constructive and informative comments of the Reviewer. Below are our point-by-point responses to the concerns raised during review process.

Point 1: Abstract should be supplemented by the results obtained from the conducted research (please add scientific and practical significance of the selected method). Rewrite it clearly stating the facts; focus more on how your research has contributed to knowledge gaps; describe research limitations for future research and restate your major findings.

Response 1: The “Abstract” was changed in accordance with the Reviewer’s comment.

Point 2: Materials and Methods used in this study are standard. The PCA analysis is not well written in the manuscript. The presented results are not sufficient and do not justify the application of this method in the paper. I suggest a complete revision of the section related to PCA analysis.

Response 2: Changed in accordance with the Reviewer’s comment. The description of the PCA analysis was improved.

Point 3: Results and Discussion: The results are partially explained and there are not all of them presented in an appropriate format.

Response 3: Changed in accordance with the Reviewer’s comment. The “Result and Discussion” were improved.

Round 2

Reviewer 2 Report

This manuscript assessed the quality parameters of grain and flour, rheological properties of dough and quality of bread prepared from flour of hybrid cultivars of wheat in comparison with population cultivars of wheat. This manuscript revealed that Hybrid wheat flour was characterized by good quality, therefore it can be used for wheat bread production. Among the hybrid cultivars, the best applicability for baking purposes characterized Hybery cv (N1, N2) due to the favourable values of the baking process parameters and bread quality.

The manuscript was revised and the author explained the significance of the study.The author also added some related research pictures to make the article more rigorous and enrich the research content.So I think this manuscript is acceptable in its current form.

Author Response

Response to Reviewer 2 Comments

The authors thank the Reviewer’s for the constructive and helpful reviews that enriched and improved the value of the manuscript.

Reviewer 3 Report

The authors added another co-author to the paper whose contribution is not clearly visible/stated.

The authors did not correct the Abstract and Materials and Methods sections according to the reviewer's instructions.

Author Response

Response to Reviewer 3 Comments

The authors thank the Reviewer’s for the constructive and helpful reviews that enriched and improved the value of the manuscript. Below are our point-by-point responses to the concerns raised during review process.

Point 1: The authors added another co-author to the paper whose contribution is not clearly visible/stated.

Response 1: Added another co-author (Assoc. prof. Ewa Szpunar-Krok) helped in preparing responses to Reviews, checking the text of the manuscript and prepared the statistical calculations and the description of the PCA analysis, which greatly facilitated the inference of research results and contributed to the improvement of the manuscript value. On line 545 in “Author Contributions” the contribution of added author is marked.

Point 2: The authors did not correct the Abstract and Materials and Methods sections according to the reviewer's instructions.

“Abstract should be supplemented by the results obtained from the conducted research (please add scientific and practical significance of the selected method). Rewrite it clearly stating the facts; focus more on how your research has contributed to knowledge gaps; describe research limitations for future research and restate your major findings.”

“Materials and Methods used in this study are standard. The PCA analysis is not well written in the manuscript. The presented results are not sufficient and do not justify the application of this method in the paper. I suggest a complete revision of the section related to PCA analysis”

Response 2: The “Abstract” (lines: 21-32) and the “Matherials and Methods” (lines: 144-147) were improved in accordance with the Reviewer’s comment.
